# Diversity and Biological Activities of Endophytic Fungi from the Flowers of the Medicinal Plant *Vernonia anthelmintica*

**DOI:** 10.3390/ijms231911935

**Published:** 2022-10-08

**Authors:** Litao Niu, Nigora Rustamova, Huxia Ning, Paiziliya Paerhati, Chunfang Lu, Abulimiti Yili

**Affiliations:** 1Key Laboratory of Plants Resources and Chemistry of Arid Zone, Xinjiang Technical Institute of Physics and Chemistry, Chinese Academy of Sciences, Urumqi 830011, China; 2State Key Laboratory Basis of Xinjiang Indigenous Medicinal Plants Resource Utilization, Xinjiang Technical Institute of Physics and Chemistry, Chinese Academy of Sciences, Urumqi 830011, China; 3Department of Biology, Samarkand State University, Samarkand 703004, Uzbekistan

**Keywords:** *Vernonia anthelmintica* flowers, biological activities, non-polar chemical composition, endophytic fungi

## Abstract

Secondary metabolites produced by endophytic fungi are an important source of biologically active compounds. The current research was focused on the biological activities of ethyl acetate extracts of fungi, isolated and identified from *Vernonia anthelmintica* flowers for the first time. In addition, an investigation of the non-polar chemical composition of dichloromethane-ethyl acetate extracts of the most active fungal strain was carried out. The isolates included *Ovatospora senegalensis* NR-03, *Chaetomium globosum* NR-04, *Thielavia subthermophila* NR-06, *Aspergillus calidoustus* NR-10, *Aspergillus keveii* XJF-23 and *Aspergillus terreus* XJF-3 species. Strains were identified by 18S rRNA gene sequencing methods and were registered in GenBank. Crude extracts of the fungi displayed in vitro biological activities, including antimicrobial and cytotoxic activities. A melanin content assay was performed on murine B16 cells. An ethyl acetate extract of *O. senegalensis* NR-03 showed high anticancer and antimicrobial activity; therefore, we also studied the non-polar chemical composition of the dichloromethane-ethyl acetate fraction and identified 52 non-polar compounds with the different medium. This investigation discovered that the secondary metabolites of the total extract of endophytic fungi could be a potential source of alternative natural antimicrobial, cytotoxic and melanin synthesis activity in their host plant, and the isolation of bioactive metabolites may provide a lead to new compounds of pharmaceutical significance.

## 1. Introduction

Endophytic fungi are an integral part of the plant’s microbiome and live inside the healthy tissues of host the plant without negative symptoms of disease [1,2,3,4]. They are also recognized as having complex associations with their host plants and other organisms, including endophytic microorganisms in their ecological niches and pathogens in the external environment [5,6]. As a significant microbial resource, endophytes exist widely in nature [7,8]. Endophytic fungi and their host plants have an intimate relationship, and fungal microorganisms can produce bioactive secondary metabolites that protect against plant pathogens and pests as a growth regulator [9]. The relationship between fungal endophytes and their host plants is significantly associated with modifications in the fungal colonization and extraction of fungal-derived pharmacologic active natural compounds [3,10,11]. The chemical contents and biological properties of the secondary metabolites synthesized by medicinal plants have been studied extensively [12,13,14]. However, the exploration of the complex biosynthesis of the metabolites of medicinal plants with their endophytic microorganisms has been insufficiently investigated. Endophytic fungi are an essential biotechnological source because they can be involved in the production of a pharmacologically active and structurally diverse range of secondary metabolites together with the host plant and might even be the sole producer [15,16,17]. Numerous bioactive natural compounds with antimicrobial [18,19], anti-inflammatory [20], cytotoxic [21,22], antioxidant [23], antidiabetic [24] and acetylcholinesterase [25] inhibitory activity have been isolated from endophytic fungi. For example, the bioactive secondary metabolites produced by the endophytic fungal strain *Aspergillus versicolor* exhibited significant antifungal activity against *Candida albicans* and *Geotrichum candidum* [26]. Likewise, metabolites obtained from the fungal endophyte *Aspergillus versicolor* F210 showed cytotoxic activity against the human cancer cell lines HL-60 [27]. The endophytic fungus *Trichoderma atroviride* was isolated from the healthy flowers of *Colquhounia coccineavar* and exhibited strong antimicrobial activity against *Staphylococcus aureus, Bacillus subtilis* and *Micrococcus luteus* [28]. Juan et al. [29] obtained secondary metabolites from the fungus *Conocybe siliginea* and evaluated the metabolites for their immunosuppressive activity against A (ConA)-induced T-cell proliferation and lipopolysaccharide (LPS)-induced B-lymphocyte-cell proliferation. The metabolites also inhibited Con A-induced T-cell proliferation. The endophytic fungus *Streptomyces* sp. HN2A53 showed satisfactory antiviral inhibitory activity against influenza virus neuraminidase (NA) [30]. A sesquiterpene coumarin produced by the endophytic fungus *Penicillium* sp. KMU18029 displayed significant cytotoxic activity against two human cancer cell lines, HL-60 and SMMC-7721, as reported by Li et al. [31]. The endophytic fungus *Peniophora incarnate* produced metabolites that demonstrate cytotoxic activity against the human A375, MCF-7 and HL-60 cancer cell lines [32]. The secondary metabolites produced by endophytic microorganisms are an important, rich source of biologically active compounds for novel drug discovery. The following characteristics were taken into consideration in order to isolate endophytic fungi from the Chinese medicinal plant *V. anthelmintica.* The endophytes of the flowers of Chinese medicinal plant *V. anthelmintica* have not been studied until now. This is the first report on the isolation of endophytic fungi from the flowers of *V. anthelmintica*. These plants have an ethno botanical history and have been used in traditional Chinese medicines for the treatment of vitiligo [33]. Moreover, natural compounds isolated and identified from different parts of *V. anthelmintica* have shown antidiabetic, anti-vitiligo, antioxidant and antimicrobial activity [34]. Endophytes may display similar biological activities and produce similar metabolites to their hosts. Therefore, we selected the medicinal plant *V. anthelmintica* grown in China for our study and isolated and systematically studied the endophytic fungi from the flowers.

In our previous work, we reported the diversity, biological activity and chemical composition of endophytic bacteria and fungi associated with roots of the Chinese medicinal plant *V. anthelmintica* [35,36,37]. The aims of the current research were (a) to isolate and identify to the endophytic fungi associated with *V. anthelmintica* flowers, (b) to measure several biological activities of the total extract, (c) to optimize the culture conditions for producing the secondary metabolites excreted by the most active fungal strain, and (d) to identify the non-polar chemical composition of the dichloromethane fraction by gas chromatography–mass spectrometry (GC–MS).

## 2. Results

### 2.1. Isolation and Identification of Endophytic Fungi from the Flowers of V. anthelmintica

In total, more than 30 fungal endophytic strains were isolated from the fresh flower tissues of the Chinese medicinal plant *V. anthelmintica*. All isolated fungal strains were cultivated in a PDB medium and extracted with ethyl acetate to produce crude extracts (Figure 1). Among these, only six crude extracts of the endophytic fungi *O. senegalensis* NR-03, *Ch. globosum* NR-04, *T. subthermophila* NR-06, *A. calidoustus* NR-10, *A. keveii* XJF-23 and, *A. terreus* XJF-3 were selected for our study. The primary screening of these crude extracts showed antimicrobial activity.

All fungal strains were identified by 18S rRNA gene sequencing and registered in GenBank (accession numbers MW996742, MW996743, MW996745, MW996749, MW881461 and MW881454 (https://www.ncbi.nlm.nih.gov/nuccore/MW881454 accessed on 24 August 2022) Figure 2). The isolated fungal strains belong to four genera: Ovatospora, Chaetomium, Thielavia and Aspergillus. The most numerous were representatives of the genus *Aspergillus* (NR-10, XJF-23 and XJF-3). The isolates showed 99–100% homology to *O. senegalensis, Ch. globosum*, *T. subthermophila, A. calidoustus*, *A. keveii* and, *A. terreus* (Table 1). A phylogenetic tree using the neighbor-joining method was constructed, showing the closest relatives of the isolates from GenBank (Figure 2).

### 2.2. Antimicrobial Activity of the Crude Extracts of Endophytic Fungi

A part of our present study focused on systematically screening crude extracts from the endophytic fungi. Our work is the first study of the antimicrobial inhibition of extracts from endophytic fungi from the flowers *V. anthelmintica*. In this regard, here, we considered the antimicrobial properties of the selected endophytic strains, namely *O. senegalensis* NR-03, *Ch. globosum* NR-04, *T. subthermophila* NR-06, *A. calidoustus* NR-10, *A. keveii* XJF-23 and *A. terreus* XJF-3. The antimicrobial properties of the crude fungal extracts were estimated using three pathogenic microbes: *E. coli* (ATCC6538) (Gram-positive bacteria), *E. coli* (ATCC11229) (Gram-negative bacteria) and *C. albicans* (ATCC10231) (fungi) (Table 2). All results were compared to the positive controls ampicillin and amphotericin B. The crude extract of *O. senegalensis* NR-03, (an ethyl acetate extract derived from the filtrate of the PDB culture) exhibited strong (zone of inhibition (ZOI) 26 and 19 mm) antibacterial activity against the *E. coli* and *S. aureus* strains, respectively, as well as weak antifungal inhibition (7.5 mm ZOI) against *C. albicans*. *Ch. globosum* NR-04 inhibited average-to-moderate activity against the pathogenic fungal strain (*C. albicans*) and bacteria (*S. aureus*), with inhibition zones of 12 mm and 20 mm, respectively (Appendix A). Three species belonging to the genus *Aspergillus,* namely, *A. calidoustus* NR-10, *A. keveii* XJF-23 *and A. terreus* XJF-3, showed the same results against *S. aureus* (ZOI 15, 15 and 16 mm, respectively). In our previous work, we reported the antimicrobial activity of endophytic fungi isolated from the roots of *V. anthelmintica.* Accordingly, two species belonging to the genus *Aspergillus* (*Aspergillus* sp. XJA6 and *A. terreus* XJA8) showed strong antifungal activity against the *C. albicans* strain with inhibition zones of 23 and 20 mm, respectively [37]. On the other hand, many species of *Aspergillus* also showed antimicrobial activity. For example: the endophytic fungi *A. candidus* NF2412 and *A. terreus* BCC51799 exhibited in vitro antimicrobial activity against a number of pathogenic microorganisms [38,39].

### 2.3. Determination of the MIC

Crude extracts of the selected fungal strains were isolated from the *V. anthelmintica* flowers. Some strains exhibited strong antibacterial activity; therefore, we also determined their of minimum inhibitory (MIC). The results were compared with the positive controls ampicillin and amphotericin B. The total extract of *O. senegalensis* NR-03 exhibited strong antibacterial action, with a MIC value of 62.5 μg/mL against *S. aureus*. *Ch. globosum* NR-04 exhibited powerful antifungal activity against the pathogenic fungus *C. albicans,* with a MIC value of 31.25 μg/mL. The strain *A. terreus* XJF-3 displayed significant activity against *E. coli* and *S. aureus,* with MIC values 62.5 and 250 μg/mL, respectively (Table 3).

### 2.4. Effect of the Crude Extracts of Endophytic Fungi Isolated from V. anthelmintica Flowers on Melanin Content and Tyrosinase Activity in B16 Cells

The host plant *V. anthelmintica* has been used on a very large scale in traditional Chinese medicine and Uyghur folk medicine [40,41]. In recent years, many bioactive components from *V. anthelmintica* have been reported, and these compounds have shown many pharmacological properties. The extract of the seeds of this plant is used in the treatment of vitiligo in traditional Uyghur medicine [33]. It has previously been reported that crude extracts of endophytic bacteria and fungi as well as secondary metabolites from the crude extracts of endophytes isolated from the roots of *V. anthelmintica* demonstrated effects on melanin synthesis in B16 melanoma cells [35,36,37]. These results suggested that the biological role of the *V. anthelmintica*-derived microbes from the aerial parts (flowers), particularly endophytes and their metabolites, remain to be investigated. The total crude extract of the ethyl acetate fraction from the endophytic fungi increased melanin synthesis in murine B16 cells in a dose-dependent manner, and all results were comparable with the drug 8-methoxypsoralen (8-MOP) (Table 4). Crude extracts of the fungal strains *O. senegalensis* NR-03, *Ch. globosum* NR-04 and *T. subthermophila* NR-06 could increase melanin synthesis in B16 cells by 154.49, 171.58 and 156.27% (50 µg/mL) compared with the positive control 8-MOP (123.34 at 50 µg/mL). The crude extracts of three species of fungi belonging to the genus *Aspergillus,* namely, *A. calidoustus* NR-10, *A. keveii* XJF-23 and *A. terreus* XJF-3, increased melanin synthesis in murine B16 cells by 193.31, 205.35 and 208.46% (50 µg/mL), respectively, which was 1.6-fold higher than that of 8-MOP (Figure 3A). Therefore, these were chosen for further investigation of their melanin synthesis and tyrosinase activity in murine B16 cells at different concentrations (Figure 3B,C). The fungal endophytes *A. calidoustus* NR-10 (5 µg/mL, 137.91%; 10 µg/mL, 162.17%; and 50 µg/mL, 197.98%), *A. keveii* XJF-23 (5 µg/mL, 96.57%; 10 µg/mL, 161.18%; and 50 µg/mL, 204.02%) and *A. terreus* XJF-3 (5 µg/mL, 133.24%; 10 µg/mL, 167.85%; and 50 µg/mL, 207.12%) showed a 1.5-fold increase in melanin synthesis compared with the positive control 8-MOP (125.52 at 50 µg/mL) in a dose-dependent manner. After melanin production was measured, the tyrosinase activities of these strains was determined in B16, as shown in Figure 3C. The secondary metabolites of three *Aspergillus* species improved the tyrosinase activity in a dose-dependent manner, and the results were compared with 8-MOP, (50 µM, 123.34%). Treatment at concentrations of 5–50 µg/mL resulted in strong tyrosinase activity. *A. calidoustus* NR-10 showed 5 µg/mL, 113.12%; 10 µg/mL, 130.02%; and 50 µg/mL, 146.63%. *A. keveii* XJF-23 showed 5 µg/mL, 95.05%; 10 µg/mL, 129.02%; and 50 µg/mL, 151.63%. *A. terreus* XJF-3 showed 5 µg/mL, 114.06%; 10 µg/mL, 140.06%; and 50 µg/mL, 146.63% (Table 5).

### 2.5. Cytotoxic Activity of Fungal Endophytes

In recent years, the natural compounds produced by endophytic fungi have become a rich resource for drug discovery. Many endophytic fungi synthesize secondary metabolites with anticancer activity. The secondary metabolites produced by the endophytic fungus *Ch. globosum* showed in vitro cytotoxicity activity against the human hepatoma cell line HepG-2 [22]. The endophytic fungus *Paramyrothecium roridum* was isolated from fresh roots of *Morinda officinalis* and demonstrated cytotoxic activity against the human SF-268, NCI-H460 and epG-2 cancer cell lines [42]. Li et al. [32] reported that the endophytic fungus *Peniophora incarnata* exhibited cytotoxic activity against the three human cancer cells lines A375, MCF-7, and HL-60. As measured in our previous work, the crude ethyl acetate extracts of the endophytic fungi *Aspergillus* sp. XJA6 and *A. terreus* XJA8 isolated from the root of *V. anthelmintica* showed strong activity against the HeLa and HT-29 cancer cell lines [37].

In our present research, the crude extracts of six endophytic fungi (*O. senegalensis* NR-03, *Ch. globosum* NR-04, *T. subthermophila* NR-06, *A. calidoustus* NR-10, *A. keveii* XJF-23 and *A. terreus* XJF-3) were selected. Crude extracts of the fungi were tested against the following human cancer cells lines: HT-29 (colon cancer), MCF-7 (breast cancer) and HeLa (cervical cancer) (Table 6). All results were compared with the positive control DOX. The crude extract of the endophytic fungus *O. senegalensis* NR-03 exhibited stronger anticancer activity than the positive control DOX against two cancer cell lines with IC_50_ values of 0.10 ± 0.004 μg/mL against HT-29 and 0.09 ± 0.005 μg/mL against HeLa. The IC_50_ values of the extract of *T. subthermophila* NR-06 displayed moderate anticancer activity against all cell lines: 3.85 ± 0.15 μg/mL against HT-29, 9.99 ± 0.69 μg/mL against MCF-7 and 5.89 ± 0.35 μg/mL against HeLa. The total extract of the endophytic fungus *A. terreus* XJF-3 showed significant cytotoxic activity against HeLa, with IC_50_ values of 0.10 ± 0.0051 μg/mL, similar to the positive control (0.11 ± 0.005 μg/mL).

### 2.6. The Effects of Different Media and Incubation Times on the Growth and Production of Metabolites from Endophytic Fungi

The bioactive secondary metabolites produced by six fungal endophytes displayed several biological activities. Among them, the *O. senegalensis* NR-03 strain exhibited significant antibacterial and cytotoxic activities. Because that strain produced more secondary metabolites than other fungal isolates, we selected it for further optimization of the growth conditions in terms of incubation time and different media content. We used different media content for growing endophytic fungi and producing bioactive secondary metabolites. SAB and PDB were the most effective media for the growth and production of secondary metabolites of the endophytic fungus *O. senegalensis* NR-03. In the SAB and PDB media, the fungal strain produced 1.98 g and 2.03 g of secondary metabolites, respectively, per liter of the liquid medium. Therefore, these media were found to be the most optimal nutrients for the growth of the fungus. The incubation time also impacted metabolite biosynthesis. The time profile of bioactive secondary metabolites synthesized by the fungal endophyte *O. senegalensis* NR-03 revealed a growth period of 42 days (Figure 4). During growth, the production of secondary metabolites was observed to increase, reaching a maximum of 2.01 g/L at 14 days. The optimal culture conditions for the growth and production of secondary metabolites of endophytic microorganisms are specific to each species. For example, the optimal conditions for the endophytic bacteria *Alternaria solani* are 30 °C for 48 h and pH 7.0 [43]. The fungal endophyte *Piriformospora indica* has optimal growth and production of secondary metabolites under the following conditions: carbon and nitrogen sources, glucose and NH_4_Cl; shaker speed, 200 rpm; working volume, 20% [44].

### 2.7. Scanning Electron Microscopy (SEM) Analysis of O. senegalensis NR-03

A scanning electron microscope is a type of electron microscope that images the samples surface by scanning it with a high-energy beam of electrons in a raster scan pattern [45]. SEM observations were carried out to determine the surface structure of the mycelia and conidiophores of the endophytic fungi. This revealed the distinct mycelia, hyphae and sporangia of the endophytic fungi (Figure 5).

### 2.8. HPLC Analysis of the Crude Extracts from Different Media

HPLC analyses were conducted for tracking the secondary metabolites of the endophytic fungus *O. senegalensis* NR-03 produced in different liquid media. All extracts of the endophytic fungus were subjected to the same conditions. HPLC analysis was performed to identify the secondary metabolites of endophytic fungus extracts from the different media as shown in Figure 6. The results showed that the different media influenced the production of metabolites by the endophytic fungi. For example, in the PBG medium, the fungus produced different metabolites (retention times: 35, 48 and 51 min), which was not observed in other media (Appendix A). *O. senegalensis* NR-03 in the SAB medium also revealed peaks which were not found for the other media between 35 and 40 min (Figure 6).

### 2.9. Chemical Composition of the Endophyic Fungus O. senegalensis NR-03 Analyzed by GC-MS

As mentioned above, we conducted several biological assays of the crude extracts of six endophytic fungi; of these, *O. senegalensis* NR-03 exhibited significant pharmacological activity. We decided to study the non-polar chemical composition of dichloromethane–ethyl acetate fractions at a ratio of 80:1 in crude extracts of *O. senegalensis* NR-03, which were separated using column chromatography. Because many non-polar compounds of medicinal plants are very important sources of a huge number of biologically active agents [46,47], our laboratory is continuing this research investigating the secondary metabolites from the ethyl acetate fraction, which have potential for future drug development.

Cell-free supernatants of fungi grown on seven different media were extracted with dichloromethane and analyzed by GC/MS. In total, 52 components were identified from the dichloromethane extract of *O. senegalensis* NR-03 (Appendix A). There are no literature data relating to the non-polar chemical composition of the endophytic fungi of *V. anthelmintica* flowers. This work was the first study investigating the non-polar components of the endophytic fungus *O. senegalensis* NR-03. In total, 16 non-polar components were identified from the extract with the YPD medium (Appendix A), including thymine (1.13%), n-acetyltyramine (2.21%) and hexahydro-3-(phenylmethyl)-pyrrolo[1,2-a] pyrazine-1,4-dione (21.64%), which were the main compounds (Table 7). Eleven components were found with the MEB medium; 3-methyl-butanoic acid (1.40%), orcinol (1.11%) and simvastatin (14.38%) were the main compounds (Appendix A). The highest number of compounds was detected with the PDB medium (23 components; 3-methyl-1,2-cyclopentanedione (1.41%), terrein (33.71%), 9,12-octadecadienoic acid (Z, Z)-methyl ester (6.39%), (E)-9-octadecenoic acid, methyl ester (6.15%) and rosenonolactone (2.11%) were the main compounds) (Appendix A). It should be noted that hexahydro-3-(phenylmethyl)-pyrrolo[1,2-a] pyrazine-1,4-dione was the main compound in almost all the liquid media.

## 3. Discussion

The Chinese medicinal plant *V. anthelmintica* is very important in traditional Chinese and Uyghur medicine. The seeds of the plant are used to treatment skin disorders, such as vitiligo, leucoderma, psoriasis and eczema [34,48]. Moreover, natural compounds isolated from the seeds and other part of *V. anthelmintica* have shown antidiabetic, anti-vitiligo, antioxidant and antimicrobial activity. In particular, an extract from the seeds is one of the most popular Uygur medicines used for vitiligo. In previous studies, we reported the diversity and synergetic properties of the total extract compositions, as well as the effects of the pure isolated secondary metabolites of the endophytic bacteria and fungi of this plant associated with the roots in terms of melanin synthesis, cytotoxic, antidiabetic, antioxidant and antimicrobial activity [35,37]. The present investigation focused on evaluating the diversity, biological activity and non-polar chemical composition of the most active endophytic fungi of *V. anthelmintica* flowers. Six endophytic fungi were isolated and identified; moreover, their antimicrobial, cytotoxic and melanin synthesis effects on murine B16 cells were evaluated. Three species belonging to the genus *Aspergillus* (*A. calidoustus* NR-10, *A. keveii* XJF-23 and *A. terreus* XJF-3) showed antimicrobial and cytotoxic activity comparable with some results reported by other authors [49,50,51]. These results revealed the extensive capacity for the production of pharmacologically active secondary metabolites with antimicrobial and cytotoxic potential from these fungal microorganisms. Fungal endophytes from the genus *Aspergillus* have been reported as important sources of bioactive natural products with applications in agriculture and pharmacology [52,53,54,55]. Recently, we isolated the endophytic bacteria and fungi from *V. anthelmintica* roots. Crude ethyl acetate extracts of the bacteria *P. ananatis* and the fungal strains *S. communae* XJA1, *Talaromyces* sp. XJA4, *Aspergillus* sp. XJA6 and *A. terreus* XJA8 greatly influenced melanin synthesis in murine B16 cells with IC_50_ values of 191.56.814, 119.02, 118.16, 182.45 and 177.92% at 50 µg/mL, respectively [35,37]. In our current work, endophytic fungi belonging to the *Aspergillus* genus (*A. calidoustus* NR-10, *A. keveii* XJF-23 and *A. terreus* XJF-3) showed a 1.6-fold increase in melanin synthesis in murine B16 cells compared with the positive control 8-MOP (123.34% at 50 µg/mL), with IC_50_ values of 193.31, 205.35 and 208.46% (50 µg/mL). The results showed that the biological activity of endophytic fungi isolated from *V. anthelmintica* flowers was higher than that of the endophytic isolates of roots. The same can be seen for cytotoxic activity. For examples, crude extracts of *Aspergillus* sp. XJA6 and *A. terreus* XJA8 isolated from the roots of *V. anthelmintica* showed strong cytotoxic activity against the HT-29 and HeLa cells lines with IC_50_ values 19.31 ± 0.8 μg/mL, 5.73 ± 0.6 μg/mL, 9.99 ± 0.8 μg/mL and 24.69 ± 0.2 μg/mL, respectively. However, *O. senegalensis* NR-03 and *A. terreus* XJF-3 displayed powerful cytotoxic activity against the HT-29 and HeLa lines, with IC_50_ values of 0.10 ± 0.004 μg/mL and 0.09 ± 0.005 μg/mL against HT-29 and 0.10 ± 0.005 μg/mL against HeLa. As shown by these results, the synergetic effect of the secondary metabolites in the total extracts very strongly increased melanin synthesis in melanoma B16 cells, as well as displaying antimicrobial and cytotoxic activity. A wide range of biological active secondary metabolites have been found from endophytic fungi. Tantapakul et al. [56] isolated a secondary metabolite from the endophytic fungus *Ch. globosum* 7s-1. The metabolite showed promising cytotoxicity in the selected cancer cell lines (KB, MCF-7, and NCI-H187) with IC50 values of 7.04, 18.40, and 0.98, respectively. Other secondary metabolite polyketides were isolated from the fungus *Phoma bellidis* and demonstrated cytotoxicity in human cancer cell lines (HL-60, A549, SMMC-7721, MCF-7, and SW480) [57]. The most active fungus (*O. senegalensis* NR-03) was further studied for optimization of the culture conditions and to determine the non-polar chemical composition of the dichloromethane-ethyl acetate fraction, to understand which secondary metabolites displayed bioactivity. The non-polar composition of endophytes is poorly understood. Endophytic fungi mainly synthesize ethers, acids, amines, alkenes and benzene [58]. Fungal strain *O. senegalensis* NR-03 synthesized different non-polar components depending on the content of the culture medium. For example, 9,12-octadecadienoic acid (Z, Z)- methyl ester and (E)-9-octadecenoic acid methyl ester were the main components in PDB the medium but were not found at all in other broth media. Contrariwise, hexahydro-3- (phenylmethyl)–pyrrolo [1,2-a] pyrazine-1,4-dione was the main component in all, except the MEB medium. Hexahydro-3-(phenylmethyl)- pyrrolo[1,2-a] pyrazine-1,4-dione (21.64%), simvastatin (14.38%), terrein (33.71%), 9,12-octadecadienoic acid (Z, Z)-, methyl ester (6.39%) and (E)-9-octadecenoic acid, methyl ester (6.15%) were the main components in different media. In this investigation, about 52 compounds were identified by GC-MS for strain *O. senegalensis* NR-03, the major compounds in the displayed biological activity. In general, these results support the development of natural products that may minimize the need for the application of pharmacology, which would be an environmentally friendly approach and preserve natural resources in medicinal chemistry. In light of the current study, further characterizations of the fungus *O. senegalensis* NR-03 will be executed in order to (a) isolate and identify the secondary metabolites from ethyl acetates extract and (b) measure the biological activity of each individually isolated natural compound.

## 4. Materials and Methods

### 4.1. Plant Samples Collection

The Chinese medicinal plant *V. anthelmintica* was collected from Hotan, within the Xinjiang Autonomous region, in August 2020. The fresh aerial parts of the plant were transported to the laboratory of Xinjiang Indigenous Medicinal Plants Resource Utilization, Xinjiang Technical Institute of Physics and Chemistry, Chinese Academy of Sciences.

### 4.2. Isolation of Endophytic Fungi Producing Bioactive Secondary Metabolites

The isolation procedures for endophytic fungi involved a surface sterilization method described in a previous publication [37]. Fresh flowers (10 g) were separated and sterilized with 99.9% ethanol for 2 min and with 10% NaClO, then rinsed with sterile distilled water. The sterile flowers (10 g fresh weight) were cut into pieces 3–4 cm long and macerated using a sterile mortar and pestle. The macerated tissue (1 g) was transferred into plastic tubes containing 9 mL of sterile phosphate buffered saline (PBS) (20 mM sodium phosphate, 150 mM NaCl, pH 7.4) and shaken for 1 min using a Vortex Biosan B-1. Aliquots (100 µL) from dilutions (10^1^–10^5^) were spread on PDA. After that, they were incubated for 7 days. Visually homogenous colonies of different sizes, shapes and colors were used for DNA isolation.

### 4.3. Identification of Endophytic Fungi Producing Bioactive Secondary Metabolites

Isolated endophytic fungi were incubated in a 250 mL flask containing 50 mL of fresh Potato Dextrose Both (PDB) medium for 16 h at 28 °C with shaking at 200 rpm. DNA extraction was conducted using an E.Z.N.A^TM^ Fungal DNA Kit (Omega Bio-Tek, USA). PCR amplification was carried out to determine the partial 18S rRNA gene. The PCR thermal cycling scheme was set as follows: 30 cycles of denaturation at 94 °C for 1 min, 55 °C for 30 s and 72 °C for 1.5 min. The 18S universal primers C27F AGAGTTTGATCCTGGCTCAG and C14921R TACGGCTACCTTGTTACGACTT were used to amplify of the 18S rRNA gene. Purification of the PCR products and determination of the sequences were performed by GeneCore BioTechnologies Co, Ltd. (Shanghai, China). The 18SrRNA gene sequence of the fungal strains obtained was compared with the NCBI database (https://www.ncbi.nlm.nih.gov/nuccore/MW881454 (accessed on 24 August 2022)). The neighbor-joining method was used to infer the phylogenetic relatedness [59]. Bootstrap analysis was performed with 500 replications [60] using the heuristic search option as described previously to estimate the reliability of the inferred monophyletic groups. The evolutionary distances were computed using the maximum composite likelihood method [61] and are given in units of the number of base substitutions per site. This analysis involved 15 nucleotide sequences and all ambiguous positions were removed for each sequence pair (pairwise deletion option). There were 767 positions in total in the final dataset. Evolutionary analyses were conducted with MEGA 6 software [62].

### 4.4. Fermentation and Crude Extract Preparation

All isolated, purified and identified endophytic fungi were cultured in a potato dextrose broth (PDB) (dextrose 20 g/L, potato extract 4 g/L) medium for the synthesis of bioactive natural compounds. The fungal culture was incubated at 28 °C on a shaker at 160 rpm for 28 days under septic conditions [63,64]. The PDB cultures were filtered by vacuum filtration and cell-free supernatants of each fungus were extracted separately with ethyl acetate (1:3) in a separator funnel. During the extraction procedure, 1–1.2 g of total extract was obtained per liter of fungal culture.

### 4.5. Antimicrobial Activity of Total Extracts of the Endophytic Fungi

The bioactive secondary metabolites produced by the endophytic fungi were screened in vitro for antibacterial and antifungal activity, which was measured using a well diffusion method [35,65]. The three pathogenic microorganisms used for the antimicrobial assays were the Gram-positive bacterium *Staphylococcus aureus* (ATCC6538), the Gram-negative bacterium *Escherichia coli* (ATCC11229) and the fungus *Candida albicans* (ATCC10231). The bacterial pathogenic strains were plated on LB medium (4% glucose, 1% peptone) and the fungus in potato dextrose agar (PDA) (4 g potato extract, 20 g dextrose, agar 20 g, and 1 L water) inoculated at 28 °C for 48 h in accordance. A total of 6 µL of crude extract of endophytic fungi dissolved in DMSO was added to these wells, with the plates being incubated at 37 °C for 24 h for bacteria and 72 h for fungi. The inhibitions zones were measured to assess the antimicrobial activity of the samples. Ampicillin was obtained from Sigma Chemicals Co., and amphotericin B was purchased from AMRESCO LLC.

### 4.6. Determination of MIC

The minimum inhibitory concentration (MIC) of the secondary metabolites of ethyl acetate extracts of the endophytic fungi were determined against *S. aureus, E. coli* and *C. albicans*. The MIC of the extracts was estimated for each of the test organisms in triplicates. Bacterial strains were cultured on the LB liquid medium (4% glucose, 1% peptone) at 37 °C for 24 h, while the fungal pathogen was grown in potato dextrose both at 30 °C for 48 h. Ampicillin and amphotericin B were used as positive controls. The total crude extracts were dissolved in DMSO, and 50 µL of each sample dilution was added in 96-well micro-titer plates, followed by the addition of 10 µL of the pathogen suspension and 100 µL of the liquid culture medium. Plates were incubated at 37 °C for 24 h. [66,67] The growth of bacteria and fungus was checked by reading the absorbance at 600 nm. The lowest concentration of the compounds that that could inhibit the growth of microorganisms was determined.

### 4.7. Melanin Content Assay

Cell Culture: Murine B16 melanoma cell lines (B16F10) were obtained from CAS (Chinese Academy of Sciences, Shanghai, China). Cells were cultured in Dulbecco’s modified Eagle’s medium (DMEM, Gibco Life Technologies, Waltham, MA, USA) supplemented with 10% heat-inactivated fetal bovine serum (FBS), penicillin G (100 U/mL), and streptomycin (100 mg/mL) (Gibco-BRL, Grand Island, NY, USA) at 37 °C in a humidified atmosphere of 5% CO_2_.

Melanin Measurement: The melanin content assay of ethyl acetate extracts of the endophytic fungi associated with *V. anthelmintica* flowers was determined in accordance with the procedure described previously [36,68,69]. Exponentially growing B16 cells were seeded at a density of 2 × 10^5^ cells per well in a 6-well plate and incubated for 24 h. After that, test samples were added to the individual wells, and the cells were incubated for 48 h, then washed twice with ice-cold PBS. After that, the cells were lysed, and the assembled cells were centrifuged at 10,000× *g* for 20 min. Each lysate (150 µL) was placed in a 96-well microplate and read spectrophotometrically at 405 nm using a multi-plate reader. The amount of protein of each sample was determined by a BCA Protein Assay Kit (Biomed, Beijing, China). The melanin content was normalized to the cellular protein concentration. Melanin content was calculated according to the formula:(1)$$Melanin\;content\;\left(\%\right)=\frac{S}{B}\times 100$$
where *S* is the absorbance of the cells treated by the samples, and B is the absorbance of the wells containing untreated cells. Each measurement was carried out in triplicate.

Tyrosinase Activity Assay: Tyrosinase activity was determined by the method of Chao et al. [68]. First, the B16 cells were seeded in a 6-well plate at a concentration of 2 × 10^5^ cells per well and incubated for 24 h. Test samples were then added to each well and incubated for 24 h. Cells were then washed twice with ice-cold PBS and lysed using 1% Triton X-100 solution containing 1% sodium deoxycholate for 30 min at −80 °C. The lysates were centrifuged at 12,000× *g* for 15 min. After quantification and adjustment of the protein concentration of the supernatants, 90 µL of the supernatant was mixed and incubated with 10 µL of a freshly prepared substrate solution (10 µM 8-MOP) in duplicate in a 96-well plate, followed by incubation at 37 °C. Samples were then read at 490 nm. The results were calculated by the formula used for calculating the melanin content.

### 4.8. Cytotoxic Activity (MTT Assay)

The cytotoxic activity of the total extracts of all fungal endophytes was tested against three human cancer cells lines: HT-29 (colon cancer), MCF-7 (breast cancer), and HeLa (cervical cancer). An MTT assay was carried out as described, with minor modifications [70,71,72]. The samples were dissolved in DMSO; the extract concentration was 50 μg/μL. The cells grown to the logarithmic growth phase were aspirated, washed once with PBS, digested with trypsin, terminated by adding a medium, gently pipetted, counted, and seeded in a 96-well plate (100 μL/well) at the corresponding cell density after, overnight culture. Samples (20 μL/well) were added and incubated in CO_2_, at 37 °C for 48 h, then, aspirated and the old medium was discarded before, adding 100 μL of MTT. The samples were left to culture for 2 h. After incubation at 37 °C for 2 h, the absorbance value (OD) at 570 nm was measured using an MB microplate reader. The inhibition rate was determined by the equation
(2)$$IR=\frac{\left(C-S\right)}{\left(C-B\right)}\times 100$$
where *C* is the OD of the control group, *S* is the OD of the experimental group, and *B* is the OD of the blank.

### 4.9. Optimization of the Optimal Culture Medium for the Production of Biologically Active Secondary Metabolites from Endophytic Fungi

The total extracts of all the isolated fungal strains were measured for their pharmacological activities (antimicrobial, cytotoxic and melanin content assay on murine B16 cells). The strain *O. senegalensis* NR-03 displayed the highest activity; in addition, the yield of biologically active compounds synthesized by this strain was significantly higher than that of the other strains. Therefore, *O. senegalensis* NR-03 was selected for further study to find the most effective culture conditions. In order to provide a way to select the best medium and incubation time for the production of the bioactive metabolites, seven different liquid culture media were used: yeast extract peptone dextrose (YPD) (yeast extract, 1 g/L; dextrose, 20 g/L; peptone, 20 g/L), malt extract broth (MEB) (malt extract 30 g/L), Sapouraud broth (SAB) (dextrose, 40 g/L; peptone, 10 g/L), Czapeck–Dox medium (CDM) (sucrose, 30 g/L; NaNO_3,_ 3 g/L; K_2_HPO_4,_ 1 g/L; MgSO_4,_ 0.5 g/L; KCl, 0.5 g/L; FeSO_4_, 0.01 g/L), beef extract broth (BEB) (beef extract, 3 g/L; NaCl, 5 g/L), peptone beef extract glycogen medium (PBG; peptone, 10 g/L; beef, 10 g/L; glycogen, 4 g/L; NaCl, 5 g/L; NaC12H25SO4, 0.1 g/L) and potato dextrose broth (PDB) (potato extract 4 g/L, dextrose 20 g/L). After incubation on the different liquid media, crude extracts were extracted with ethyl acetate. The total extract of the endophytic fungi was tested for biological activity and analyzed by HPLC.

### 4.10. The Effect of Cultivation Time on the Secondary Metabolite Production of Endophytic Fungi

The time profile of the secondary metabolite production by the endophytes was observed a for growth period of 42 days. Every 7 days, the fungal culture was adjusted to 7, 14, 21, 28, 35 and 42 days, respectively. The extracts were obtained with ethyl acetate, and the yield of the metabolites produced was calculated.

### 4.11. Scanning Electron Microscopy Analysis of Endophytic Fungi

The mycelia of endophytic fungi were placed in a petri dish on the surface of a PDA medium as mycelial plug and cultured for 14 days. The mycelia of the fungi were directly coated with gold by an iron sputter coater (SuPro, ISC 150, Bruker Nano GmbH, Berlin, Germany), and the appearance of the surface of the fungi, such as shape, color, the diameter of the colony and colony reversal, was observed under a scanning electron microscope (Carl Zeiss Jena, SUPRA 55VP, Bruker Nano GmbH, Berlin, Germany).

### 4.12. HPLC Analysis

Crude extracts of the endophytes (15 mg) were dissolved in 1 mL of HPLC grade methanol, and HPLC was carried out with the DIONEX Ultimate 3000 HPLC system (Thermo-Fisher, Waltham, MA, USA) combined with a Sunfire C18 column (4.6 × 250 mm, 5 µm) (Waters, USA) and a Sunfire C18 guard column (4.6 × 20 mm, 5 µm) (Waters, Waltham, MA, USA). Mobile phase A was 0.2% (*v*/*v*) formic acid (HCOOH) in water; mobile phase B was acetonitrile (ACN). The elution profile was 0–10 min 100% B (isocratic), and 1–86 min 10–100% B (acetonitrile). The flow rate was 1 mL/min, and the injection volume was 10 μL. UV chromatograms were recorded at 210, 254 and 330 nm; this analysis enabled the characterization of alkaloids, terpenoids and phenolic compounds on the basis of their retention time and UV spectra.

### 4.13. Gas Chromatography-Mass Spectrometry (GC-MS) Analysis

The fungal strain was incubated in seven different liquid media: yeast extract peptone dextrose (YPD), malt extract broth (MEB), cornmeal broth medium (CBM), Sabouraud broth (SAB), Czapeck–Dox medium (CDM), beef extract broth (BEB), peptone beef extract glycogen medium (PBG) and potato dextrose broth (PDB). The secondary metabolites produced by the fungal strain in each medium were extracted by ethyl acetate then dissolved in dichloromethane. The non-polar chemical composition of the endophytic fungus *O. senegalensis* NR-03 was subjected to gas chromatography—mass spectral (GC-MS) analysis, as previously reported [37,58]. The GC-MS system consisted of an Agilent 7890 gas chromatograph coupled to an Agilent 5975 mass detector (Agilent Technologies, New York, USA) and an Agilent 7693 autosampler. A capillary column HP-5 (30 m × 0.25 mm internal diameter; CM Scientific, New York, USA), coated with a 0.25-µM film of 5% (*v*/*v*) dimethyl polysiloxane and 95% diphenyl was used for separation. The column temperature was set at 100 °C and held for 5 min. A sample of 0.5 µL of the solutions (5 mg extract was dissolved in 1 mL dichloromethane) was injected [73,74]. The initial column temperature was increased to 145 °C at a rate of 5 °C/min and was held at this temperature for 25 min. The MS inlet temperature was 250 °C; MS mode, EI; mass range, 40–550 amu; and ionization source temperature, 280 °C. The compounds were identified on the basis of retention time (RT) and by comparing the spectra with a stored MS library (W8N05ST and NIST08).

### 4.14. Statistical Analysis

All the experimental data were analyzed using GraphPad Prism using three replicate values in side-by-side subcolumn analysis of variance (ANOVA) and Tukey’s test to determine statistical significance. A *p*-value of <0.05 was considered to be statistically significant. The correlation index was calculated using the Pearson coefficient (ρ).

## 5. Conclusions

In this study, we demonstrated the biological activities of the endophytic fungi associated with Chinese medicinal plant *V. anthelmintica* flowers. Six fungal strains were studied; three isolates belonged to the *Aspergillus* genus and the others were from *Ovatospora*, *Chaetomium* and *Thielavia*. Crude extracts of endophytic fungi exhibited several biological effects, including antimicrobial, cytotoxic and melanin syntheses and tyrosinase activity, on murine B16 cells. The strain *O. senegalensis* NR-03 displayed strong antibacterial activity against the *E. coli* and *S. aureus* strains. Moreover, the crude extract of the endophytic fungus *O. senegalensis* NR-03 exhibited stronger anticancer activity than the positive control DOX against two cancer cell lines HT-29 and HeLa. *O. senegalensis* NR-03 showed higher biological activity than the other fungal strains. Bioactive compounds, which were produced by endophytic fungi *O. senegalensis* NR-03, were analyzed and determined through GC-MS analyses. In addition, we observed optimization of the growth conditions in terms of incubation time and different media content. The secondary metabolites produced by most active fungal strains showed cytotoxicity against several cancer cell lines, in addition to displaying antimicrobial and melanin synthesis activity, which suggest that it should be considered as a drug candidate with therapeutic potential.

## Figures and Tables

**Figure 1 ijms-23-11935-f001:**
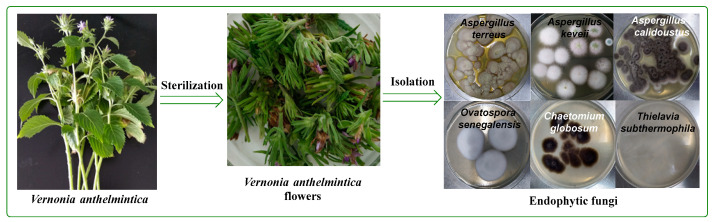
Isolated fungal strains from the flowers of the medicinal plant *V. anthelmintica*.

**Figure 2 ijms-23-11935-f002:**
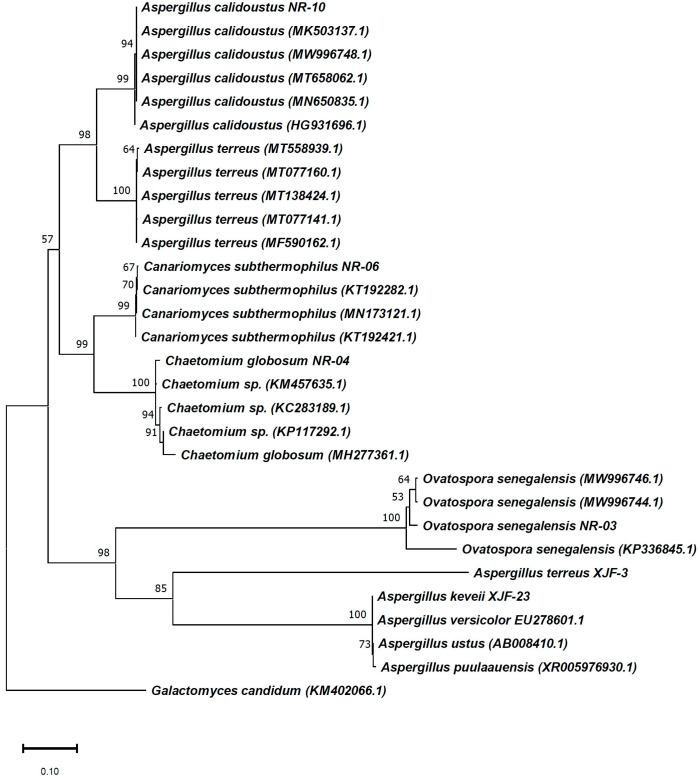
The phylogenetic tree of endophytic fungi isolated from *V. anthelmintica*, with the closest relatives registered in GenBank of NCBI.

**Figure 3 ijms-23-11935-f003:**
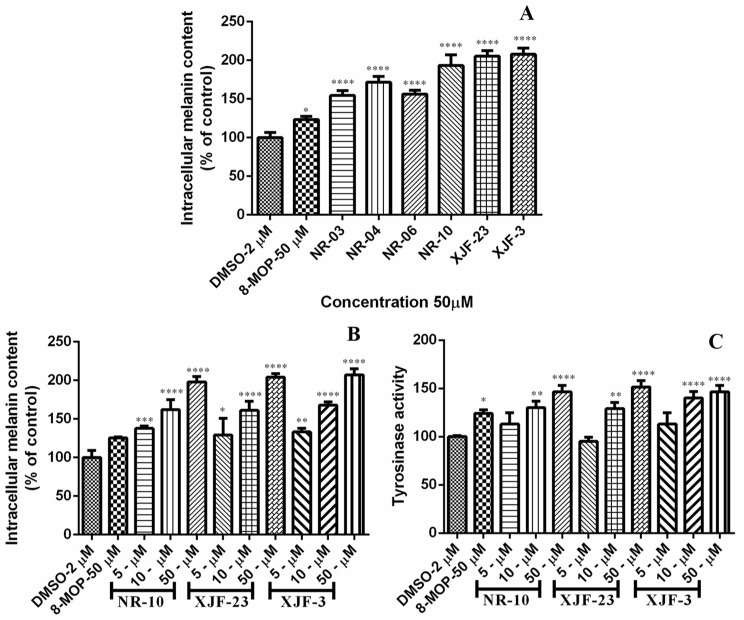
(**A**) Stimulation of melanin content of B16 cells by secondary metabolites of crude extract of the fungi; (**B**) melanin content activity of different concentration; (**C**) tyrosinase activity of different concentration. Note: * compared to the blank control group (DMSO), *p* < 0.05; ** compared to the blank control group (DMSO), *** *p* < 0.001 compared with the blank control group (DMSO); *p* < 0.01 **** compared to the blank control group (DMSO), *p* < 0.0001.

**Figure 4 ijms-23-11935-f004:**
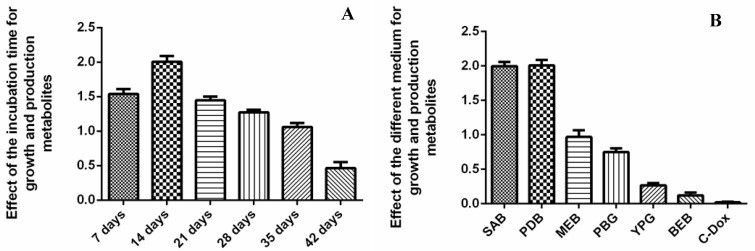
Optimization of incubation time (**A**) and different medium for production secondary metabolites (**B**) by leading endophytic fungus of *O. senegalensis* NR-03.

**Figure 5 ijms-23-11935-f005:**
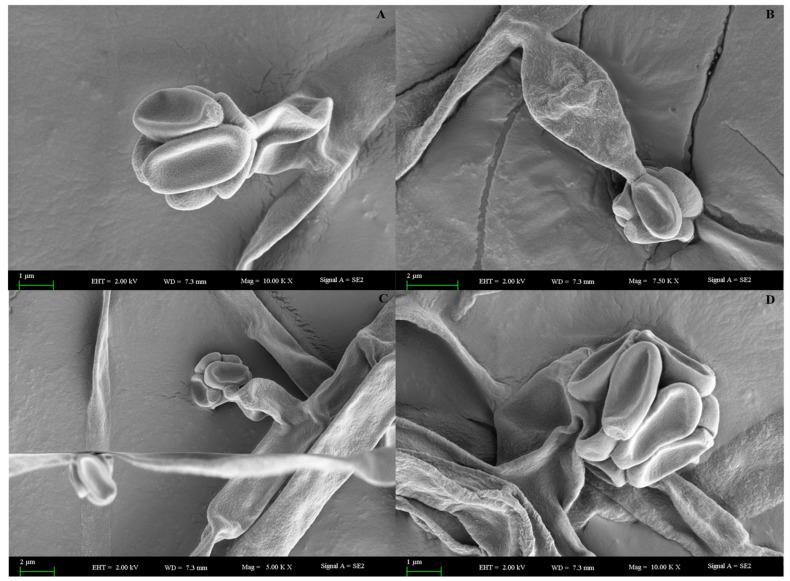
Microscopical view of *O. senegalensis* NR-03 conidial heads ×1000, ×750 (**A**,**B**) and (**C**,**D**) hyphae ×500, mycelium ×1000 and conidiophore.

**Figure 6 ijms-23-11935-f006:**
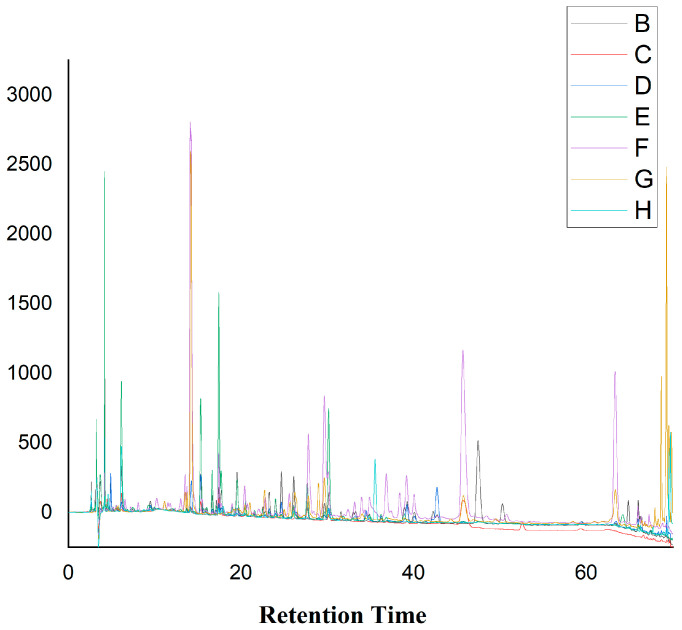
Analytical HPLC chromatograms of the crude extract of endophytic fungus from different medium. Recorded at 254 nm and performed using a waters column 412 and acetonitrile, 0.1% acidified water with formic acid as mobile phase. B—PBG; C—Czapeck-Dox; D—BEB; E—YPD; F—SAB; G—MEB; H—PDB.

**Table 1 ijms-23-11935-t001:** Sequence similarities of endophyte fungi isolated from aerial parts of *V. anthelmintica* with sequences registered in GenBank.

Isolated Strains Sequences Deposited in GenBank		Closest Match among Bacteria (18S rRNA Genes) (GenBank)		
Strains	Accession Number	Species	Source	ID%
NR-03	MW996742	*O. senegalensis*	flower	100
NR-04	MW996743	*Ch. globosum*	flower	99.04
NR-06	MW996745	*T. subthermophila*	flower	100
NR-10	MW996749	*A. calidoustus*	flower	100
XJF-23	MW881461	*A. keveii*	flower	99.82
XJF-3	MW881454	*A. terreus*	flower	99.94

**Table 2 ijms-23-11935-t002:** Antimicrobial activity (ZOI) of endophytic fungi associated with medicinal plant *V. anthelmintica* flowers.

Sample	Sample Concentration	Sample Amount (μL)	*C. albicans* (mm ZOI)	*E. coli* (mm ZOI)	*S. aureus* (mm ZOI)
Ampicillin	312.5 μg/mL	20	-	11	-
Ampicillin	4.44 μg/mL	20	-	-	18.5
Amphotericin B	5 μg/mL	20	12	-	-
*O. senegalensis* NR-03	50 μg/mL	20	7.5	26	19
*Ch. Globosum* NR-04	50 μg/mL	20	12	-	20
*T. Subthermophila* NR-06	50 μg/mL	20	9.5	9	11
*A. calidoustus* NR-10	50 μg/mL	20	10	13	15
*A. keveii* XJF-23	50 μg/mL	20	8	10	15
*A. terreus* XJF-3	50 μg/mL	20	10	11	16

**Table 3 ijms-23-11935-t003:** Determination of minimum inhibitory (MIC) concentration of endophytic fungi associated with medicinal plant *V. anthelmintica* flowers.

Sample	*C. albicans*	*E. coli*	*S. aureus*
	MIC μg/mL	MIC μg/mL	MIC μg/mL
*O. senegalensis* NR-03	ND	2000	62.5
*Ch. Globosum* NR-04	31.25	ND	125
*T. Subthermophila* NR-06	ND	ND	ND
*A. calidoustus* NR-10	ND	125	ND
*A. keveii* XJF-23	ND	250	ND
*A. terreus* XJF-3	250	62.5	250
Ampicillin 100 mg/mL	-	0.98	62.5
Amphotericin B 50 mg/mL	0.98	-	-

**Table 4 ijms-23-11935-t004:** Stimulation of melanin content of B16 cells by total crude extracts of the endophytic fungi.

Sample	Concentration (µM)	Intracellular Melanin Content (% of Control)
Control	2µL-DMSO	100.00
8-MOP	50μM	123.34
*O. senegalensis* NR-03	50μg/mL	154.49
*Ch. globosum* NR-04	50μg/mL	171.58
*T. subthermophila* NR-06	50μg/mL	156.27
*A. calidoustus* NR-10	50μg/mL	193.31
*A. keveii* XJF-23	50μg/mL	205.35
*A. terreus* XJF-3	50μg/mL	208.46

**Table 5 ijms-23-11935-t005:** Effect of crude extracts of endophytic fungi on melanin production and tyrosinase activity in B16 cells.

Sample	Concentration	Intracellular Melanin Content (% of Control)	Tyrosinase Activity
Control	DMSO	100.00	100.00
8-MOP	50 µM	125.52	124.8
*A. calidoustus* NR-10	5 µM	137.91	113.12
	10 µM	162.17	130.02
	50 µM	197.98	146.63
*A. keveii* XJF-23	5 µM	96.57	95.05
	10 µM	161.18	129.02
	50 µM	204.02	151.63
*A. terreus* XJF-3	5 µM	133.24	114.06
	10 µM	167.85	140.06
	50 µM	207.12	146.63

**Table 6 ijms-23-11935-t006:** Cytotoxic activity of total extracts obtained from fungal endophytes on human cancer cell lines HT-29, MCF-7 and HeLa (*n* = 3).

Samples	Cell Lines		
	IC_50_ (μg/mL)		
	HT-29	MCF-7	HeLa
*O. senegalensis* NR-03	0.10 μg/mL ± 0.004	Not active	0.09 μg/mL ± 0.005
*Ch. Globosum* NR-04	32.41 μg/mL ± 2.20	Not active	29.38 μg/mL ± 1.27
*T. Subthermophila* NR-06	3.85 μg/mL ± 0.15	9.99 μg/mL ± 0.69	5.89 μg/mL ± 0.35
*A. calidoustus* NR-10	16.44 μg/mL ± 0.85	19.55 μg/mL ± 1.03	13.59 μg/mL ± 0.63
*A. keveii* XJF-23	31.37 μg/mL ± 1.82	Not active	17.81 μg/mL ± 0.78
*A. terreus* XJF-3	Not active	Not active	0.10 μg/mL ± 0.005
DOX	0.82 μg/mL ± 0.041	0.17 μg/mL ± 0.006	0.11 μg/mL ± 0.005

**Table 7 ijms-23-11935-t007:** The chemical composition of dichloromethane-ethyl acetate of the *O. senegalensis* NR-03.

№	Composition	t_R_ (min)	Relative Peak Area %						
			BEB	SAB	YPD	MEB	PBG	CDM	PDB
1	3-Methyl-butanoic acid	3.418	-	-	-	1.40	-	-	-
2	2-Methyl-butanoic acid	3.520	-	-	-	0.48	-	-	0.78
3	p-Xylene	3.741	-	-	-	-	-	1.71	-
4	3-methyl-1,2-cyclopentanedione	5.848	-	-	-	-	-	-	1.41
5	2-Methyl-phenol	6.247	-	0.21	-	0.12	-	-	0.22
6	p-Cresol	6.553	-	1.96	-	0.28	-	-	-
7	Tetramethyl-pyrazine	6.714	-	-	0.26	-	-	-	-
8	Maltol	7.105	-	-	-	0.57	-	-	-
9	Phenylethyl alcohol	7.139	-	-	0.13	-	-	-	-
10	N-(3-Methylbutyl)acetamide	7.402	-	-	0.16	-	-	-	-
11	Benzeneacetic acid	9.136	-	3.02	-	-	-	-	1.00
12	Indole	9.756	-	-	0.56	-	-	-	-
13	Orcinol	10.801	-	1.66	-	1.11	-	-	-
14	4-Hydroxy-2-methylbenzaldehyde	11.157	-	-	-	-	-	-	1.11
15	3,4-Bis(methylene)-cyclopentanone	11.480	-	-	0.36	-	-	-	-
16	4-Hydroxy-benzeneethanol	11.489	-	-	-	0.99	-	-	-
17	3-Ethyl-2,5-dimethyl-pyrazine	11.795	-	-	0.53	-	-	-	-
18	Terrein	11.948	-	-	0.92	-	-	-	33.71
19	3,5-Diethyl-2-methyl-pyrazine	12.474	-	-	0.30	-	-	-	-
20	Dimethyl(3-methylphenoxy)ethoxy-silane	12.525	-	-	-	-	-	-	1.09
21	N-(2-Phenylethyl)-acetamide	12.542	-	-	0.76	-	-	-	-
22	Thymine	13.587	-	-	1.13	-	-	-	-
23	3-Mercaptobenzoic acid, S-methyl-, methyl ester	14.258	-	-	-	0.44	-	-	-
24	Tetradecanoic acid	15.278	-	-	-	-	-	-	0.22
25	Hexahydro-pyrrolo[1,2-a]pyrazine-1,4-dione	15.337	-	-	0.41	-	-	-	-
26	Pentadecanoic acid	16.008	-	0.17	-	-	-	-	1.10
27	N,N-Dimethyl-1H-purin-6-amine	16.153	-	-	0.30	-	-	-	-
28	N-Acetyltyramine	16.280	-	-	2.21	-	-	-	-
29	Hexadecanoic acid, methyl ester	17.223	-	-	-	-	-	-	1.94
30	Hexahydro-3-(2-methylpropyl)-pyrrolo[1,2-a]pyrazine-1,4-dione	17.393	36.59	-	-	-	-	7.90	-
31	3,4-dihydro-6,8-dihydroxy-3-methyl-1H-2-benzopyran-1-one	17.546	-	-	-	0.44	-	-	-
32	n-Hexadecanoic acid	17.656	-	-	-	-	-	-	0.60
33	Eicosamethyl-cyclodecasiloxane,	17.903	-	-	-	-	-	-	0.31
34	Hexadecanoic acid, ethyl ester	18.081	-	-	-	-	-	-	0.34
35	9,12-Octadecadienoic acid (Z,Z)-, methyl ester	19.449	-	-	-	-	-	-	6.39
36	(E)-9-Octadecenoic acid, methyl ester	19.525	-	-	-	-	-	-	6.15
37	Methyl stearate	19.848	-	-	-	-	-	-	0.87
38	Linoleic acid ethyl ester	20.358	-	-	-	-	-	-	0.78
39	(E)-9-Octadecenoic acid ethyl ester	20.434	-	-	-	-	-	-	0.62
40	N-[2-(1H-indol-3-yl)ethyl]-acetamide	20.859	-	-	0.76	-	-	-	-
41	3-Benzyl-6-isopropyl-2,5-piperazinedione	21.845	-	-	0.22	-	-	-	-
42	Octadecamethyl-cyclononasiloxane	21.989	-	-	-	-	-	-	0.36
43	Hexadecane	22.244	-	-	-	-	-	3.05	
44	Hexahydro-3-(phenylmethyl)-pyrrolo[1,2-a]pyrazine-1,4-dione	22.915	14.44	1.47	21.64	-	14.47	18.13	0.56
45	Tetracosane	23.671	-	-	-	-	-	4.89	-
46	Eicosane	25.098	-	-	-	-	-	5.60	-
47	Rosenonolactone	26.152	-	-	-	-	-	-	2.11
48	(5.beta.)-chol-7-ene	27.129	-	-	-	0.81	-	-	-
49	1,8-dihydroxy-3-methoxy-6-methyl-9,10 anthracenedione	27.256	-	-	-	0.22	-	-	-
50	3,4-dihydro-3-hydroxy-2,2-dimethyl-, (R)-2H-naphtho[1,2-b]pyran-5,6-dione	31.087	-	-	-	-	0.83	-	-
51	Simvastatin	31.391	-	-	-	14.38	15.68	-	1.60
52	Lovastatin	34.307	-	-	-	-	-	-	0.67

## Data Availability

The data presented in this study are available in the article and Appendix A.

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
