# Peer review of "Diversity and Biological Activities of Endophytic Fungi from the Flowers of the Medicinal Plant Vernonia anthelmintica"

_ijms, 2022, doi:10.3390/ijms231911935_

Round 1

Reviewer 1 Report

The article describes the study of six endophytic fungi from the flowers of Vernonia anthelmintica. The ethyl acetate extracts were obtained, and a screening of the in vitro antimicrobial and cytotoxic activities were performed. The extract of the fungus O. senegalensis NR-03 showed good activities, but the bioactive compound was not isolated. Then, I suggest isolation of the compound responsible for the activity. Even if the ethyl acetate extract was the most active anticancer, why was the composition of the volatiles of the dichloromethane extract studied? It was unclear how the volatile compounds could be responsible for the activities shown by the extracts. Also, some figures in SM are too small and not visible. Thus, despite the good results presented in the biological activities studied, I do not recommend the article for publication in IJMS.

Author Response

REVIEWER 1.

Comment 1.

The article describes the study of six endophytic fungi from the flowers of Vernonia anthelmintica. The ethyl acetate extracts were obtained, and a screening of the in vitro antimicrobial and cytotoxic activities were performed. The extract of the fungus O. senegalensis NR-03 showed good activities, but the bioactive compound was not isolated. Then, I suggest isolation of the compound responsible for the activity.

Response: We thank the referee for these comments and suggestion. Our laboratory is continuing to pursue this research and our research team isolated individual secondary metabolites from ethyl acetate extract of the fungus O. senegalensis NR-03, to understand exactly which compounds in the total extract has been shown biological activity.

-------------------------------------------------------------------------------------------------

Commnet 2.  

Even if the ethyl acetate extract was the most active anticancer, why was the composition of the volatiles of the dichloromethane extract studied? It was unclear how the volatile compounds could be responsible for the activities shown by the extracts.

Response: Thank you for your comments.  Many non-polar compounds of medicinal plants a very important sources a huge number of biologically active agents. Therefore, we decided to study of non-polar chemical composition of endophtic fungus which isolated from Chinese medicinal plant V. anthelmintica. we added some sentences in the section 2.8 about this. In the future we will investigated chemical composition of ethyl acetate extract of O. senegalensis NR-03

------------------------------------------------------------------------------------------------

Comment 3.

Also, some figures in SEM are too small and not visible. Thus, despite the good results presented in the biological activities studied, I do not recommend the article for publication in IJMS.

Response: We have improved SEM of the O. senegalensis NR-03 (Figure 5), The number of images has been reduced and their size has been increased.

Reviewer 2 Report

The work protocol is properly structured, but the presentation of the methods does not reveal all the test stages.
For example, when testing the antimicrobial activity, the extracts and the positive control were evaluated, but not the solvents (ethyl acetate or DMSO). At least it doesn't appear from the text. Studies indicate that above a certain concentration, both have antimicrobial activity.
Regarding the additional material, I don't understand what L2, L6, S4 are...

Working methods are not fully presented. For example, 4.6 talks about determining the MIC, but not the MBC.

The text, starting with the title, must be proofread (the English translation is not the most appropriate; measure units : milliliters instead of microliters).

Author Response

REVIEWER 2.

Comment 1.

The work protocol is properly structured, but the presentation of the methods does not reveal all the test stages. For example, when testing the antimicrobial activity, the extracts and the positive control were evaluated, but not the solvents (ethyl acetate or DMSO). At least it doesn't appear from the text. Studies indicate that above a certain concentration, both have antimicrobial activity.

Response: Thank you for your comments. Based on your several useful suggestions, we have added some sentences of the methods, for reveal all the test stages. Regarding antimicrobial assay of the crude extract of endophytic fungi, we have used Ampicillin and Amphotericin B as positive control. We couldn't have used negative control (DMSO and ethyl acetate).

-------------------------------------------------------------------------------------------------

Comment 2.

Regarding the additional material, I don't understand what L2, L6, S4 are...

Response: Has been changed L2, L6, S4 to name of the strains.

-------------------------------------------------------------------------------------------------

Comment 3.

Working methods are not fully presented. For example, 4.6 talks about determining the MIC, but not the MBC.

The text, starting with the title, must be proofread (the English translation is not the most appropriate; measure units: milliliters instead of microliters).

Response: Thank you for your comments. In our revised manuscript, we performed a major revision of the English language (grammar, verbs and other errors) using an English Editing Service (MDPI), which has improved our manuscript. Moreover, we have deleted data of MBC from the table 3.

Reviewer 3 Report

The aim of the current manuscript was to study the diversity and biological activates of endophytic fungi from the medicinal plant of Vernonia anthelmintica flowers. The study presents results of some importance.

Following comments are made for improvement.

In the title correct “activates”. Is it “activities”?

The abstract is not compact and deficient in some aspects.

Introduction is good and provides background for the conducted studies but requires language editing. Some of the sentences have not been properly structured and do not make sense.
The methodologies and protocols used are adequate but lack consistency. These have been described in adequate detail.

In the section 4.1, Selection of plant material, provide information in a continuous paragraph instead of points.

Results although presented in detail but requires some revision to clarify some of the statements. The results have not been discussed properly. It is advisable to discuss the results with the findings of other researchers.

The conclusions are good however; the authors are advised to make some recommendation on the basis of their results.

The standard of English is not up to the mark. The manuscript is full of syntactical errors. There is no proper use of verbs, tenses, words, articles on numerous occasions. There were many cases in the manuscript where the sentences were not properly structured, suggesting that the paper would improve if it could be edited for language and technical correctness.

In some cases, the authors used first form of speech. Usually results are written in the third form of speech with passive voice.

Some of the references do not conform to the style of the journal.

Double-check all the citations in the text to ensure that they are listed in the references list and vice versa.

Author Response

REVIEWER 3.

Comment 1.

The aim of the current manuscript was to study the diversity and biological activates of endophytic fungi from the medicinal plant of Vernonia anthelmintica flowers. The study presents result of some importance.

Following comments are made for improvement.

Response: We thank for the time they spent analysing this manuscript. We are also very grateful for the helpful comments from you.  

-------------------------------------------------------------------------------------------------

Comment 2.

In the title correct “activates”. Is it “activities”?

Response: Corrected to “activities”

-------------------------------------------------------------------------------------------------

Comment 3.

The abstract is not compact and deficient in some aspects.

Response:  Thank you for your comment, we have added some sentence in the abstract.

-------------------------------------------------------------------------------------------------

Comment 4.

Introduction is good and provides background for the conducted studies but requires language editing. Some of the sentences have not been properly structured and do not make sense.

Response: Thank you for your comments. In our revised manuscript, we performed a major revision of the English language (grammar, verbs and other errors) using an English Editing Service (MDPI), which has improved our manuscript.

-------------------------------------------------------------------------------------------------

Comment 5.

The methodologies and protocols used are adequate but lack consistency. These have been described in adequate detail.

Response: Thank you for your comments. Based on your several useful suggestions, we have added some sentences of the methods, for reveal all the test stages.

-------------------------------------------------------------------------------------------------

Comment 6.

In the section 4.1, Selection of plant material, provide information in a continuous paragraph instead of points.

Response: Has been corrected.

-------------------------------------------------------------------------------------------------

Comment 7.

Results although presented in detail but requires some revision to clarify some of the statements. The results have not been discussed properly. It is advisable to discuss the results with the findings of other researchers.

Response: Thank you for your comments and suggestions. Based on your several useful suggestions, we have discussed the results with other authors reports.

-------------------------------------------------------------------------------------------------

Comment 8.

The conclusions are good however; the authors are advised to make some recommendation on the basis of their results.

Response: Has been added some recommendation, with your suggestions.

-------------------------------------------------------------------------------------------------

Comment 9.

The standard of English is not up to the mark. The manuscript is full of syntactical errors. There is no proper use of verbs, tenses, words, articles on numerous occasions. There were many cases in the manuscript where the sentences were not properly structured, suggesting that the paper would improve if it could be edited for language and technical correctness. In some cases, the authors used first form of speech. Usually results are written in the third form of speech with passive voice.

Response: Thank you for your comments. In our revised manuscript, we performed a major revision of the English language (grammar, verbs and other errors) using an English Editing Service (MDPI), which has improved our manuscript.

-------------------------------------------------------------------------------------------------

Comment 10.

Some of the references do not conform to the style of the journal.

Double-check all the citations in the text to ensure that they are listed in the references list and vice versa.

Response: All references improved to the style of the journal.

Reviewer 4 Report

Major revision needed

1-      Rewrite abstract to show the aim of the work and present the results in a good way.

2-      Introduction is not comprehensive for the topic of the study, please rewrite introduction section to become more show importance of this study and support it with recent references

3-      Check all microbial strains, make it italic

4-      Check all abbreviations, please write it complete in the first time, then write it abbreviated

5-      Unify all unites in one style and check all sub- and superscript numbers

6-      Write in details crude extraction preparation method

7-      Write in details antimicrobial activity method

8-      Line 374, change ml to µl

9-      Line 378, add how to read fungi

10-  From where you obtain Ampicillin and Amphotericine B

11-  How ampicillin 4.44 µg/ml gave activity on S. aureus but ampicillin 312.5 µg/ml did not give activity?

12-  Are you used  Amphotericine B in concentration 5 or 50?

13-  In MIC test (Table 3), how MBC lower than MIC in O. senegalensis NR-03???????????????????

14-  Give a discussion on inactivity of some fungal extracts on MCF7

15-  In figure 7, add unit on vertical axes

16-  Phytochemical analysis is required.

17-    Add these two studies to support this sentence with recent reference ‘’’ Fungal endophytic from the ge- 280 nus Aspergillus have been reported as important sources of bioactive natural products 281 with applications in agriculture and pharmacology’’’’’  https://link.springer.com/article/10.1007/s12010-022-03876-x        https://link.springer.com/article/10.1007/s12010-021-03702-w

18-  Line 283, change entophytic to endophytic

19-  Many confusing expressions can be seen in the manuscript, so authors should revise carefully.

Author Response

REVIEWER 4.

Comment 1. Rewrite abstract to show the aim of the work and present the results in a good way.

Response: Thank you again for your useful comment. In our revised manuscript and we have added some sentences based on your suggestions. The main of the work is clearly described at the end of the introduction.

-------------------------------------------------------------------------------------------------

Comment 2. Introduction is not comprehensive for the topic of the study, please rewrite introduction section to become more show importance of this study and support it with recent references

Response: Thank you for your comment and suggestion. We revised the introduction section and added some sentence

-------------------------------------------------------------------------------------------------

Comment 3. Check all microbial strains, make it italic

Response: Corrected

-------------------------------------------------------------------------------------------------

Comment 4. Check all abbreviations, please write it complete in the first time, then write it abbreviated

Response: Corrected

-------------------------------------------------------------------------------------------------

Comment 5. Unify all unites in one style and check all sub- and superscript numbers

Response: Corrected

-------------------------------------------------------------------------------------------------

Comment 6. Write in details crude extraction preparation method

Response: We have written details in the section of “4.4. Fermentation and crude extracts preparation” and marked with red color in the text.

-------------------------------------------------------------------------------------------------

Comment 7. Write in details antimicrobial activity method

Response: Thank you for your comment. We have written details in the section of 4.5. Antimicrobial activity of total extract of endophytic fungi.

-------------------------------------------------------------------------------------------------

Comment 8. Line 374, change ml to µl

Response: Has been changed.

-------------------------------------------------------------------------------------------------

Comment 9. Line 378, add how to read fungi

Response: Pathogenic bacteria and fungus also reading the absorbance 600 nm, in the text also improved.

-------------------------------------------------------------------------------------------------

Comment 10. From where you obtain Ampicillin and Amphotericine B

Response: Added in section 4.5. Antimicrobial activity of total extract of endophytic fungi

-------------------------------------------------------------------------------------------------

Comment 11. How ampicillin 4.44 µg/ml gave activity on S. aureus but ampicillin 312.5 µg/ml did not give activity?

Response: Thank you for your comments. We used ampicillin in two different doses for two pathogenic bacteria. For E. coli 312.5 µg/ml and for S. aureus 4.44 µg/ml respectively.

-------------------------------------------------------------------------------------------------

Comment 12. Are you used Amphotericine B in concentration 5 or 50?

Response: Here in Amphotericine B in concentration is 5 μg/mL

-------------------------------------------------------------------------------------------------

Comment 13. In MIC test (Table 3), how MBC lower than MIC in O. senegalensis NR-03???????????????????

Response: Thank you for your comments. In our revised manuscript and we have deleted data of MBC from the Table 3.

-------------------------------------------------------------------------------------------------

Comment 14. Give a discussion on inactivity of some fungal extracts on MCF7

Response: Thanks for your comment again. We have added in discussion part some reports others authors, cytotoxic activity (on human cancer cell MCF7) of secondary metabolites isolated from other endophtic fungi.

-------------------------------------------------------------------------------------------------

Comment 15. In figure 7, add unit on vertical axes

Response: Thank you for comment. Here is not figure 7, maybe figure 6 about HPLC analyses of crude extract of most active fungus. We cannot analyze this data, because it was taken a long time ago and now has been deleted from the computer's memory. If you don't want this Figure to remain, we can remove it from the next revision stages of our manuscript.

-------------------------------------------------------------------------------------------------

Comment 16. Phytochemical analysis is required.

Response: Thank you very much for your comment. But in the present study focused only, the dichloromethane-ethyl acetate (80:1) fraction of the most active fungus analyzed by GCMS. Here in we have obtained first fraction (dichloromethane-ethyl acetate 80;1) from ethyl acetate extract of fungus separated by column chromatography method. We are currently continuing this work in our laboratory, isolating secondary metabolites individually from subsequent fractions.

-------------------------------------------------------------------------------------------------

Comment 17. Add these two studies to support this sentence with recent reference ‘’’ Fungal endophytic from the ge- 280 nus Aspergillus have been reported as important sources of bioactive natural products 281 with applications in agriculture and pharmacology’’’’’ https://link.springer.com/article/10.1007/s12010-022-03876-x        https://link.springer.com/article/10.1007/s12010-021-03702-w

Response: We have cited, based your suggestion references.

-------------------------------------------------------------------------------------------------

Comment 18. Line 283, change entophytic to endophytic

Response: Has been changed

-------------------------------------------------------------------------------------------------

Comment 19. Many confusing expressions can be seen in the manuscript, so authors should revise carefully.

Response: Thank you for your comments. In our revised manuscript, we performed a major revision of the English language (grammar, verbs and other errors) using an English Editing Service (MDPI), which has improved our manuscript.

Reviewer 5 Report

Dear authors.

Below you will find the observations/suggestions/changes to be made to the manuscript:

1) At the end of the introduction, the objective of the work is stated, but when reading it, there are several objectives. So, is it a general objective or are they specific objectives?

2) It improve the resolution of images (Figure 6), include axis units, reference and column dimensions.

3) There are two tables 5 (line 169 and 195), which is which?

4) Line 86: Satisfactory activity? How do you know that the activity is bad, acceptable, good or satisfactory?

5) Line 114: Strong antimicrobial activity? against what do you compare to know what is strong? or what scale do you use?

6) Line 121: Table 2 says antimicrobial activity, which one ZOI?, MIC?, MBC?. Please indicate which one? Remember that a table or figure should inform and not generate questions.

7) Line 169: Effect of secondary metabolitesm, which ones?

8) Line 195-196: What is the value of n=?

9) Linea 321: "Chinese medicinal plant V. anthelmintica's endophytis did not study until now". In the article Microorganisms 2020, 8(4): 586 it says it is the first study, so what is the first?

10) Line: 322: "This is a first report towards an isolation of endophytic fungi and bacteria from the aerial parts (stems, leaves, flowers and seeds) of V. anthelmintica herb". The title of this manuscript only mentions flowers, also in line 72, so what did they do?

11) Line 321-334: Why does it appear in methods and not in introduction?

12) Line 336-337: Isolation procedures for endophytic fungi we have used surface sterilisition method 336 described in a previous publication [43]. Reference 43 corresponds to a review in Phytochem Rev (2020) 19:425-448. Authors should be careful when writing a manuscript, be clear, coherent and precise.

13) What is the methodology used? Authors should write in detail all the methodology and methods used in this work.

14) Lines 243, 247, 250, 464: these lines mention apolar compounds. The authors should explain why they used this expression, if polar metabolites are registered in the tables.

15) Authors should include a paragraph at the end of the discussion indicating the strengths and weaknesses of the work done.

16) All chromatography/MS results must carry/indicate/write the percentage match with the database. Which column was used?, column dimensions?, operating mode of the furnace?, operating conditions of the MS?, etc. Must use the corresponding scientific notation for chromatography. The methodology should be rewritten and updated.

17) The conclusions correspond to the results found under one methodology. They are not perspectives. Please rewrite the section

Authors make the requested changes and update the manuscript.

Regards

Reviewer

Author Response

REVIEWER 5.

Dear authors.

Below you will find the observations/suggestions/changes to be made to the manuscript:

Author: Thank you for your comments, suggestions, thanks for the time you spent analysing this manuscript.

-------------------------------------------------------------------------------------------------

Comment 1. At the end of the introduction, the objective of the work is stated, but when reading it, there are several objectives. So, is it a general objective or are they specific objectives?

Response: Thank you for your comment. As mentioned above, at the end of the introduction we have writing the general aims of the work are. The aims of our work is not limited to these report, our work continues, our future we have planned to isolate individual compounds from the ethyl acetate extract of the most active endophytic fungus. This makes it possible to understand exactly which compound exhibits biological activity.

-------------------------------------------------------------------------------------------------

Comment 2. It improve the resolution of images (Figure 6), include axis units, reference and column dimensions.

Response: Thank you for comment. Here is not figure 7, maybe figure 6 about HPLC analyses of crude extract of most active fungus. We cannot analyze this data, because it was taken a long time ago and now has been deleted from the computer's memory. If you don't want this Figure to remain, we can remove it from the next revision stages of our manuscript.

-------------------------------------------------------------------------------------------------

Comment 3. There are two tables 5 (line 169 and 195), which is which?

Response: Line 169 Table 5 “Effect of crude extracts of endophytic fungi on melanin production and tyrosinase activity in B16 cells”. Line 195 Table 6. “Cytotoxic activity of total extracts obtained from fungal endophytes on human cancer cell lines HT-29, MCF7 and HeLa”.

-------------------------------------------------------------------------------------------------

Comment 4. Line 86: Satisfactory activity? How do you know that the activity is bad, acceptable, good or satisfactory?

Response: Crude extract of six fungal strains showed from moderate to strong antimicrobial activity. This sentence has been corrected in the text as well.

-------------------------------------------------------------------------------------------------

Comment 5. Line 114: Strong antimicrobial activity? against what do you compare to know what is strong? or what scale do you use?

Response: Thanks for your comment. The all results compared to the positive control as Ampicillin and Amphotericin B.

-------------------------------------------------------------------------------------------------

Comment 6. Line 121: Table 2 says antimicrobial activity, which one ZOI?, MIC?, MBC?. Please indicate which one? Remember that a table or figure should inform and not generate questions.

Response: In the Table 2. determined of the inhibitions zone the antimicrobial activity.

-------------------------------------------------------------------------------------------------

Comment 7. Line 169: Effect of secondary metabolitesm, which ones?

Response: Thanks you for your useful comment. Corrected caption of Table 6 based on your comment to "Table 5. Effect of crude extracts of endophytic fungi on melanin production and tyrosinase activity in B16 cells."

-------------------------------------------------------------------------------------------------

Comment 8. Line 195-196: What is the value of n=?

Response: Cytotoxic activity of endophytic fungi measured by IC50 (μg/mL)

-------------------------------------------------------------------------------------------------

Comment 9. Linea 321: "Chinese medicinal plant V. anthelmintica's endophytis did not study until now". In the article Microorganisms 2020, 8(4): 586 it says it is the first study, so what is the first?

Response:  Thanks for your comment. The present work focused to study endophytic fungi associated with V. anthelmintica flowers for firs time. These inaccuracies have been corrected in the text

-------------------------------------------------------------------------------------------------

Comment 10. Line: 322: "This is a first report towards an isolation of endophytic fungi and bacteria from the aerial parts (stems, leaves, flowers and seeds) of V. anthelmintica herb". The title of this manuscript only mentions flowers, also in line 72, so what did they do?

Response:  Thank you for your comment. We have revised in the text, this manuscript described endophytic fungi from the flowers of V. anthelmintica

-------------------------------------------------------------------------------------------------

Comment 11. Line 321-334: Why does it appear in methods and not in introduction?

Response: Thank you for your comment. We added this section in introduction. And we changed to 4.1. Plant samples collection.

-------------------------------------------------------------------------------------------------

Comment 12. Line 336-337: Isolation procedures for endophytic fungi we have used surface sterilisition method 336 described in a previous publication [43]. Reference 43 corresponds to a review in Phytochem Rev (2020) 19:425-448. Authors should be careful when writing a manuscript, be clear, coherent and precise.

Response: Thank you again for your useful comment. In our revised manuscript, changed reference.

-------------------------------------------------------------------------------------------------

Comment 13. What is the methodology used? Authors should write in detail all the methodology and methods used in this work.

Response: Thank you again for your useful comment. In our revised manuscript and we have rewritten the methods details

-------------------------------------------------------------------------------------------------

Comment 14. Lines 243, 247, 250, 464: these lines mention apolar compounds. The authors should explain why they used this expression, if polar metabolites are registered in the tables.

Response: Although there is no strict definition, usually essential oil components obtained as a result of hydro/steam distillation, sorption on a wire, or other acceptable method, are called as volatile compounds. In our case, the term "non-polar" is considered correct. Therefore, in the text, we changed “volatile” to "non-polar"

-------------------------------------------------------------------------------------------------

Comment 15. Authors should include a paragraph at the end of the discussion indicating the strengths and weaknesses of the work done.

Response: Thank you for your comment again. Based on your useful suggestions, we have added some recommendation.

-------------------------------------------------------------------------------------------------

Comment 16. All chromatography/MS results must carry/indicate/write the percentage match with the database. Which column was used? column dimensions?, operating mode of the furnace?, operating conditions of the MS?, etc. Must use the corresponding scientific notation for chromatography. The methodology should be rewritten and updated.

Response: Thanks for your useful comment. We have been writing details which column we have used, operating conditions of the MS.

-------------------------------------------------------------------------------------------------

Comment 17 The conclusions correspond to the results found under one methodology. They are not perspectives. Please rewrite the section

Authors make the requested changes and update the manuscript.

Response: Thank you for your comment. We have rewritten the section conclusions.

-------------------------------------------------------------------------------------------------

Round 2

Author Response

We are very happy to know that the reviewer rated our paper highly

Reviewer 2 Report

This study offers encouraging results to develop alternative methods that can work when traditional therapies fail.

Author Response

(The authors gave the same response as above.)

Reviewer 4 Report

Accept

Author Response

(The authors gave the same response as above.)

Reviewer 5 Report

Dear Authors.

Thank you for your responses. Here are my comments and observations:

1) When you submit a response on the question/comment/change made in your manuscript, you should inform in which line(s) the changes were made.

2) Reading some of your answers I observe/notice/analyse that they are not correct. Let me give you a precise advice.

3) The dichloromethane fraction is not apolar, so the extracted compounds are not apolar. Because the solvent dichloromethane is the aprotic polar solvent. Aprotic solvents are solvents that do not contain acidic hydrogens, so they cannot form hydrogen bonds. The value of the dielectric constant is about 9.1; the value of the dipole moment is 1.60 debyes. Dichloromethane (DCM) is a polar molecule because it has polar bonds. The polarity of a molecule is the result of the vectorial sum of all the polarities of all the bonds present in a molecule and also of the contribution of the free electrons. In conclusion: the DCM solvent is not non-polar.

Similarly, the solvent ethyl acetate (EtOAc) is not a non-polar solvent. The value of the dielectric constant is about 6.0.

The solvents DMC and EtOAc are classified as having medium polarity. Therefore, the compounds extracted with these solvents (DCM and EtOAc) have characteristics of medium polarity. This is evidenced by the extracted compounds such as acids, phenols, aldehydes, alcohols and others, which are recorded in table 6.

Therefore, you must make the changes in the manuscript. It is not an option, it is a must.

4)  Line 96 (formerly line 86) "Primary screening of these crude extracts 95 showed moderate to strong antimicrobial activity" : For antimicrobial activity, when speaking in quantitative or semi-quantitative terms, a reference (previous work or articles) should be included for comparison and indicate whether it has low, moderate or high antimicrobial activity.  Above how many millimetres (mm) in the zone of inhibition (ZOI) it is considered moderately high, in what mm range of ZOI it is considered moderate activity and below how many mm it is considered low/slow/no activity.

You should include the reference scales in order to be able to classify antimicrobial activity as high, moderate, low or no antimicrobial activity.

5) Line 224-225 (formerly 195-196). The number of times the test was performed (n= 2, 3, 4 ... ?) must be included in order to be able to associate a standard deviation value. The question was specific: what was the value of n=?, and include it in the title, heading, etc. The same applies to all other tables or figures where applicable.

6) Comment 14 made to version 1 of the manuscript relates directly to what is explained in comment 2 of this review/evaluation of your manuscript. The term apolar is not correct and does not apply to the present work.

You must make the changes.

7) Comment 15 made to version 1 of the manuscript says strengths and weaknesses.  I cannot find the update/change(s) made in the manuscript. 

Your answer is not concrete (recommendations?), because it does not evade the request/recommendation/suggestion given for the update of your manuscript.

8) Line 542 "... mass spectra library with > 70 searching ..." - I don't understand what you mean.

In comment 16 to version 1 of the manuscript, a request was made to include a column in table 6, including the database match rate of the reported compound. It was also requested to use the scientific notation in chromatography for the retention time acronym which is tR (min). 

For illustration/clarification please see Table 5 of Article: PhOL 2016, 2(1): 28-37.  Biossay-guided study in leaves of Pentacalia nitida (Baskin) and Pentacalia corymbosa (Romerillo). https://pharmacologyonline.silae.it/files/archives/2016/vol2/PhOL_2016_2_A005_Sequeda_28_37.pdf

You must make the requested changes.

9) Lines 549-563 (Conclusions): This section starts with the word in summary (?). It is not a summary, not perspectives or recommendations, not activities to be carried out in the future. It is the conclusions! To conclude is to report findings or discoveries that provide new information on the topic. 

This section should be rewritten. 

Authors should make the requested changes to the manuscript, without evading responsibility for doing so. 

Dear authors, please make changes to the manuscript and submit the updated version.

Regards

Reviewer

Author Response

RESPONSE TO REVIEWER COMMENTS

REVIEWER 5.

Dear Authors.

Thank you for your responses. Here are my comments and observations:

Author: We thank the referee for these comments.

Comment 1. When you submit a response on the question/comment/change made in your manuscript, you should inform in which line(s) the changes were made.

Response: Thank you for your comment. Based on your several useful suggestions, we have informed in which line has been changed.

----------------------------------------------------------------------------------------------------

Comment 2. Reading some of your answers I observe/notice/analyse that they are not correct. Let me give you a precise advice.

Response: Thank you for your advice.

----------------------------------------------------------------------------------------------------

Comment 3. The dichloromethane fraction is not apolar, so the extracted compounds are not apolar. Because the solvent dichloromethane is the aprotic polar solvent. Aprotic solvents are solvents that do not contain acidic hydrogens, so they cannot form hydrogen bonds. The value of the dielectric constant is about 9.1; the value of the dipole moment is 1.60 debyes. Dichloromethane (DCM) is a polar molecule because it has polar bonds. The polarity of a molecule is the result of the vectorial sum of all the polarities of all the bonds present in a molecule and also of the contribution of the free electrons. In conclusion: the DCM solvent is not non-polar.

Similarly, the solvent ethyl acetate (EtOAc) is not a non-polar solvent. The value of the dielectric constant is about 6.0.

The solvents DMC and EtOAc are classified as having medium polarity. Therefore, the compounds extracted with these solvents (DCM and EtOAc) have characteristics of medium polarity. This is evidenced by the extracted compounds such as acids, phenols, aldehydes, alcohols and others, which are recorded in table 6.

Therefore, you must make the changes in the manuscript. It is not an option, it is a must.

Response: Thank you for your advice. Based on your useful suggestions and comments, we have revised our manuscript

----------------------------------------------------------------------------------------------------

Comment 4. Line 96 (formerly line 86) "Primary screening of these crude extracts 95 showed moderate to strong antimicrobial activity" : For antimicrobial activity, when speaking in quantitative or semi-quantitative terms, a reference (previous work or articles) should be included for comparison and indicate whether it has low, moderate or high antimicrobial activity.  Above how many millimetres (mm) in the zone of inhibition (ZOI) it is considered moderately high, in what mm range of ZOI it is considered moderate activity and below how many mm it is considered low/slow/no activity.

You should include the reference scales in order to be able to classify antimicrobial activity as high, moderate, low or no antimicrobial activity.

Response: Thank you for your comment. We have revised our manuscript line 95 and 96. Here described why we have selected for our study only six fungal strains, among 30. The antimicrobial activity of the six fungi selected by us is described in detail in the section "2.2. Antimicrobial activity of the crude extracts of endophytic fungi".

The antibacterial action was classified following the Rota et al. scale:

ZOI ≤ 12 mm-weak activity

ZOI ranging between >12 and <20 mm- moderate activity

ZOI ≥ 20 mm- strong activity

Based on the scale above we have corrected the reference scales in order to be able to classify antimicrobial activity as high, moderate, low antimicrobial activity. (Line 126,127 and 134) in section 2.2.

----------------------------------------------------------------------------------------------------

Comment 5. Line 224-225. The number of times the test was performed (n= 2, 3, 4 ... ?) must be included in order to be able to associate a standard deviation value. The question was specific: what was the value of n=?, and include it in the title, heading, etc. The same applies to all other tables or figures where applicable.

Response: Each result of bioexperiment was repeated for 3 independent experiments (n= 3). Data were analyzed using the GraphPad Prism software (version 7.0) and the values are presented as the mean ± standard deviation (mean ±SD) (Line 224-225).

----------------------------------------------------------------------------------------------------

Comment 6. Comment 14 made to version 1 of the manuscript relates directly to what is explained in comment 2 of this review/evaluation of your manuscript. The term apolar is not correct and does not apply to the present work.

You must make the changes.

Response: Thank you for your comment. In the manuscript text we have used the term non-polar, not apolar.

----------------------------------------------------------------------------------------------------

Comment 7. Comment 15 made to version 1 of the manuscript says strengths and weaknesses.  I cannot find the update/change(s) made in the manuscript. 

Your answer is not concrete (recommendations?), because it does not evade the request/recommendation/suggestion given for the update of your manuscript.

Response:  Thank you for your comments. Based on your useful comment, we have added (Line 362-370) a paragraph at the end of the discussion indicating the strengths and weaknesses of the work done.

----------------------------------------------------------------------------------------------------

Comment 8. Line 542 "... mass spectra library with > 70 searching ..." - I don't understand what you mean.

In comment 16 to version 1 of the manuscript, a request was made to include a column in table 6, including the database match rate of the reported compound. It was also requested to use the scientific notation in chromatography for the retention time acronym which is tR (min). 

For illustration/clarification please see Table 5 of Article: PhOL 2016, 2(1): 28-37.  Biossay-guided study in leaves of Pentacalia nitida (Baskin) and Pentacalia corymbosa (Romerillo). https://pharmacologyonline.silae.it/files/archives/2016/vol2/PhOL_2016_2_A005_Sequeda_28_37.pdf

You must make the requested changes.

Response: Thank you for your comment. We have revised our manuscript based on your comments. On the Table 7 we have changed tR (min). Line 546-547 also corrected. “In comment 16 to version 1 of the manuscript, a request was made to include a column in table 6, including the database match rate of the reported compound”. I`m sorry I can`t understand what you mean.

----------------------------------------------------------------------------------------------------

Comment 11. Lines 549-563 (Conclusions): This section starts with the word in summary (?). It is not a summary, not perspectives or recommendations, not activities to be carried out in the future. It is the conclusions! To conclude is to report findings or discoveries that provide new information on the topic. 

This section should be rewritten. 

Authors should make the requested changes to the manuscript, without evading responsibility for doing so. 

Dear authors, please make changes to the manuscript and submit the updated version.

Response: Thank you for your comment. We have rewritten conclusions section (Line 554-566).
